# The Dissolution Behavior of Pyrite and Chalcopyrite During Low-Temperature Pressure Oxidation: Chalcopyrite Influence on Pyrite Oxidation

**DOI:** 10.3390/ma17205132

**Published:** 2024-10-21

**Authors:** Kirill Karimov, Maksim Tretiak, Denis Rogozhnikov, Oleg Dizer

**Affiliations:** Laboratory of Advanced Technologies in Non-Ferrous and Ferrous Metals Raw Materials Processing, Institute of New Materials and Technologies, Ural Federal University, Yekaterinburg 620002, Russia; kirill_karimov07@mail.ru (K.K.); darogozhnikov@yandex.ru (D.R.); oleg.dizer@yandex.ru (O.D.)

**Keywords:** pressure leaching, pyrite, chalcopyrite, sulfuric acid, pressure oxidation, sulfide, elemental sulfur, electrochemical couple

## Abstract

The research of this paper was carried out on the low-temperature (100 ± 2 °C) pressure (0.2–0.8 MPa) leaching of pyrite, chalcopyrite and their mixture. According to experiments on chalcopyrite dissolution, increasing the oxygen pressure from 0.2 up to 0.8 MPa had a slight effect on chalcopyrite dissolution. Oxygen pressure and initial sulfuric acid concentration in the range of 10–50 g/L had the greatest positive effect on the pyrite oxidation. The SEM and EDX mappings indicate the chalcopyrite and pyrite surfaces to be passivated by elemental sulfur. The oxidation degree of pyrite in its mixture with chalcopyrite increased significantly from 54.5 up to 80.3% in 0–240 min. The reaction time is relative to the dissolution of the individual mineral, while the dissolution of chalcopyrite remained virtually unchanged. The addition of Cu (II) and Fe (III) ions does not influence pyrite dissolution when chalcopyrite is added in a leaching process, which can be explained by the formation of an electrochemical link between the minerals. The positive effect of chalcopyrite addition is associated with a decreased formation of elemental sulfur on the surface of pyrite. The described method can be used for the hydrometallurgical processing of copper raw materials with increased pyrite content, as well as for the pretreatment of copper concentrates with gold-rich pyrite concentrates to increase the recovery of gold and silver.

## 1. Introduction

The main direction of the development of hydrometallurgical methods is conducting the process at low temperatures (100 ± 2 °C) in an autoclave or even under atmospheric pressure (0–0.8 MPa), which could significantly reduce both the costs of reagents and energy resources (sulfur oxidation to its elemental form rather than to sulfate), and those of expensive equipment. However, the problem of passivation of the mineral surface by reaction products arises during elemental sulfur formation, which leads to a sharp decrease in the rate of the process. To intensify leaching processes under atmospheric and pressure conditions, mechanical activation methods, the use of surfactants, catalysts and high pressures are widely used, which make it possible to disintegrate the sulfide matrix, reduce the thickness or even eliminate the formation of passivating films, which allows for increasing the rate and completeness of the opening of minerals during subsequent leaching with solutions of sulfuric and nitric acids, as well as their mixtures [1,2,3,4,5,6,7,8].

To process off-balance and hard-to-process ores, the method of bacterial leaching is also widely used [9,10,11,12]. The leach solution percolates through the material bulk or is mixed with it in stirred reactors; bacteria oxidize sulfide minerals, resulting in the surface of precious metals becoming available for oxidation at subsequent stages of processing. Significant disadvantages of such bacterial leaching include low rate and completeness of oxidation, as well as increased requirements for maintaining the vital activity of bacteria—during both the technological process and storage periods. Processes based on ultrafine grinding [13,14,15,16] allow for raising the process rate not only by expanding the reaction surface but it has also been proven that with particle sizes of less than 9 μm and a large number of pores, the thickness of the resulting films can no longer prevent leaching.

Due to this, the extraction degree of valuable components in the solution increases up to 95–99%, depending on the raw material. However, the issues of high operating costs at the grinding stage remain unresolved since, despite the development of energy-efficient mills such as IsaMill™ (Glencore Technology Pty Ltd., Brisbane, Australia), the energy costs still reach 60% or more of the cost of these processes.

Today, there are various theories regarding the occurrence of the passivation layer formed during the leaching of sulfide minerals; the nature and mechanisms of its formation have become the subject of a number of research and review papers [17,18,19,20,21]. The following passivating films are presented in various works: elemental sulfur, both in the form of porous and compact films; metal-depleted sulfides; polysulfides and iron precipitates, such as jarosite KFe_3_(SO_4_)_2_(OH)_6_ and ferrihydrite Fe_5_O_3_(OH)_9_. It is assumed that the type and structure of the films are greatly influenced by the type of mineral and leaching conditions: acid concentration, redox potential, temperature and reaction time. However, this issue has not yet been fully studied.

Studying the conditions of the formation of certain passivating films and their structure by various scientific groups has made it possible to develop several leaching methods that can be carried out at low temperatures with no use of ultrafine grinding (less than 9 μm). The first group refers to the use of surfactants, which lead to sulfur granule destruction, which prevents sulfur from passivating sulfide particles. e.g., AAC/UBC technology [22] involves pressure leaching at a temperature of 150 °C of a chalcopyrite concentrate pre-ground to 20 μm in the presence of surfactants. The copper extraction degree under these conditions is more than 95% for 2 h.

It was later found that the addition of pyrite during ferrous sulfate leaching significantly increased copper extraction [23]. This process, known as Galvanox™, proceeds at atmospheric pressure and temperature, which eliminates the high cost of an autoclave required for high-temperature pressurized processes. In addition, since the operating temperature is low and the chemical conditions are mild, an almost quantitative yield of elemental sulfur is observed. There is no need to use surfactants or ultrafine grinding. FeS_2_ and CuFeS_2_ are assumed to form a galvanic couple, where FeS_2_ acts as an additional surface for the reduction of Fe^3+^ ions, thereby increasing the rate of anodic dissolution of CuFeS_2_. Subsequently [24,25], it was revealed that the catalytic effect of adding pyrite strongly depends on the quality of the pyrite itself, and most importantly on the silver content therein.

Nazari et al. [25] suggested that the net effect of silver in the chalcopyrite leaching process is an increased electrical conductivity of the elemental sulfur layer and that the high rate of chalcopyrite dissolution is due to the galvanic interaction between pyrite and chalcopyrite. They later concluded that the fast leaching kinetics observed in silver-treated pyrite in the Galvanox™ process could not be achieved with silver ions alone since the process also required the presence of pyrite (i.e., to form a galvanic cell). However, the works of other authors [26,27,28,29] previously showed that compact sulfur is formed in the absence of a catalyst, and a porous sulfur layer is formed on the surface after the addition of silver, which leads to an increased rate of the process.

Ghahreman et al. [30,31], using XPS analysis, have shown that silver addition increases the diffusion rate of copper ions through the passivation layer by creating vacancies in the metal-depleted sulfide surface layer.

As described above, there is still no consensus regarding the mechanism of action of various catalysts on the electrochemical dissolution of metals from sulfides and their mutual influence on each other. Moreover, most studies have been aimed at analyzing chalcopyrite dissolution in ferrous sulfate solutions. Much of the research has focused on chalcopyrite dissolution in the presence of pyrite under atmospheric conditions. The behavior of these minerals at higher temperatures under pressure conditions has been poorly studied.

Thus, the vector of advanced research in the field of opening hard-to-process sulfide raw materials is directed towards improving hydrometallurgical processes. The technologies of soft autoclaves and atmospheric openings are of greatest interest. The use of these methods will reduce the consumption of energy and material resources.

## 2. Materials and Methods

### 2.1. Analysis

Chemical analysis of the starting minerals and the resulting solid dissolution products was carried out using an ARL Advant’X 4200 wavelength dispersive spectrometer (Thermo Fisher Scientific Inc., Waltham, MA, USA). Phase analysis was performed on an XRD 7000 Maxima diffractometer (Shimadzu Corp., Tokyo, Japan).

Particle sizes were analyzed using laser diffraction using an Analysette 22 Nanotec Plus instrument (FRITSCH GmbH, Idar-Oberstein, Germany).

Chemical analysis of the resulting solutions was carried out by inductively coupled plasma mass spectrometry (ICP-MS) on an Elan 9000 instrument (PerkinElmer Inc., Waltham, MA, USA).

Scanning electron microscopy (SEM) was performed using a JSM-6390LV microscope (JEOL Ltd., Tokyo, Japan) equipped with a module for energy-dispersive X-ray spectroscopy analysis (EDX).

### 2.2. Materials and Reagents

Natural sulfide minerals were used as the main raw materials: chalcopyrite (Vorontsovskoe deposit, Sverdlovsk region, Jekaterinburg, Russia) and pyrite (Berezovskoe deposit, Sverdlovsk region, Jekaterinburg, Russia), and their X-ray diffraction patterns are presented in Figure 1. All minerals used were crushed and sieved on laboratory sieves and a working fraction with particle sizes of 80% class ≤ 40 μm was selected after sifting; the granulometric composition of the minerals is presented in Figure 2. The chemical composition of the minerals used is presented in Table 1. All other reagents used were of analytical grade.

### 2.3. Equipment and Experimental Technique

Laboratory experiments on pressure leaching were carried out on a titanium reactor with a volume of 1.0 L (Parr Instrument, Moline, IL, USA), with the ability to supply and regulate oxygen flow using a flowmeter (Bronkhorst EL-FLOW Prestige and Bronkhorst EL-PRESS Metal-Sealed pressure regulators (Bronkhorst High-Tech B.V., Ruurlo, The Netherlands)) with temperature control. Stirring was carried out using a top-drive mixer to ensure pulp homogeneity.

Before the experiment, a pulp was prepared from sulfide minerals (20 g) weighed on an analytical balance and a 600 cm^3^ solution containing H_2_SO_4_, Fe_2_(SO_4_)_3_ and CuSO_4_ at certain concentrations. The reactor filling factor was 0.6. After loading the pulp, the reactor was sealed, the mixer was started and the pulp was heated up to the required temperature of 100 °C. The rotation speed of the mixer was maintained at 800 rpm, ensuring uniform pulp density. When the set temperature was reached, the reaction gas (oxygen) was supplied and the start of the experiment was recorded. At the end of the experiment, the oxygen supply was stopped and the reactor was cooled down to 70 °C. The pulp was filtered and the cake was washed and dried to constant weight. Samples for analysis were taken from liquid and solid products.

The dissolution degree of sulfide minerals in the mixture was calculated using the following method:
The mass of copper in the concentrate was calculated using Equation (1):(1)$$m_{\mathrm{Cu}(\mathrm{conc})}=\frac{{\%\mathrm{Cu}}_{\mathrm{conc}}\times m_{\mathrm{sample}}}{100},$$
where %Cu_conc_ is the percentage of copper in the concentrate, m_sample_ is the mass of the sample;The mass of copper in the cake was calculated using Equation (2):(2)$$m_{\mathrm{Cu}(\mathrm{cake})}=\frac{{\%\mathrm{Cu}}_{\mathrm{cake}}\times m_{\mathrm{cake}}}{100},$$
where %Cu_cake_ is the percentage of copper in the cake, m_cake_ is the weight of the cake;The mass of iron in the concentrate was calculated using Equation (3):(3)$$m_{\mathrm{Fe}(\mathrm{conc})}=\frac{{\%\mathrm{Fe}}_{\mathrm{ct}}\times m_{\mathrm{sample}}}{100},$$
where %Fe_ct_ is the percentage of iron in the pyrite concentrate, m_sample_ is the mass of the sample;The mass of iron in the cake was calculated using Equation (4):(4)$$m_{\mathrm{Fe}(\mathrm{cake})}=\frac{{\%\mathrm{Fe}}_{\mathrm{cake}}\times m_{\mathrm{cake}}}{100},$$
where %Fe_cake_ is the percentage of iron in the cake, m_cake_ is the mass of the cake;The percentage of chalcopyrite oxidation from the concentrate was calculated using Equation (5):(5)$$\varepsilon_{Cu}=100-\frac{{\%\mathrm{Cu}}_{\mathrm{cake}}\times m_{\mathrm{cake}}}{{\%\mathrm{Cu}}_{\mathrm{conc}}\times m_{\mathrm{sample}}}\times 100,$$
where %Cu_conc_ is the percentage of copper in the concentrate, m_sample_ is the mass of the sample, %Cu_cake_ is the percentage of copper in the cake, m_cake_ is the weight of the cake;The percentage of pyrite oxidation was calculated using Equation (6):(6)$$\varepsilon_{Fe}=100-\frac{{\%\mathrm{Fe}}_{\mathrm{cake}}\times m_{\mathrm{cake}}}{{\%\mathrm{Fe}}_{\mathrm{ct}}\times m_{\mathrm{sample}}}\times 100$$
where %Fe_ct_ is the percentage of iron in the concentrate, m_sample_ is the mass of the sample, %Fe_cake_ is the percentage of iron in the cake, m_cake_ is the weight of the cake.

## 3. Results

We studied the influence of the oxygen partial pressure (0.2–0.8 MPa); the initial concentration of sulfuric acid (10–50 g/L), copper (II) ions (1–3 g/L) and iron (III) ions (2–10 g/L); and the duration of experiments (10–240 min) on the dissolution of individual chalcopyrite, pyrite and their mixture. The temperature of the experiments was constant (100 °C).

### 3.1. Dissolution of Chalcopyrite

The interaction of chalcopyrite with sulfuric acid in the presence of oxygen and iron (III) ions may occur through the following reactions:2CuFeS_2_ + 2H_2_SO_4_ + 7O_2_ = 2CuSO_4_ + Fe_2_(SO_4_)_3_ + S^0^ + 2H_2_O ∆G (100 °C) = −2204.168 kJ/mol(7)
CuFeS_2_ + H_2_SO_4_ + 2.5O_2_ = CuSO_4_ + FeSO_4_ + S^0^ + H_2_O ∆G (100 °C) = −809.056 kJ/mol(8)
CuFeS_2_ + 2Fe_2_(SO_4_)_3_ = CuSO_4_ + 5FeSO_4_ + 2S^0^ ∆G (100 °C) = −70.752 kJ/mol(9)
CuFeS_2_ + 4O_2_ = CuSO_4_ + FeSO_4_ ∆G (100 °C) = −1238.377 kJ/mol(10)
4FeSO_4_ + O_2_(g) + 2H_2_SO_4_ = 2Fe_2_(SO_4_)_3_ + 2H_2_O ∆G (100 °C) = −304.494 kJ/mol(11)

According to the presented reactions, chalcopyrite sulfide sulfur may be oxidized by oxygen to form elemental sulfur and sulfate ions (Reactions (7)–(9)). Iron (III) ions may also act as an oxidizing agent, with sulfide sulfur being converted to an elemental one (Reaction (10)) and iron (II) ions reacting with oxygen to form iron (III) ions (Reaction (11)).

The influence of oxygen partial pressure; the initial concentration of sulfuric acid, copper (II) and iron (III) ions; and the duration on the dissolution of chalcopyrite are presented in Figure 3.

According to Figure 3a, increasing the oxygen pressure from 0.2 up to 0.8 MPa had little effect on chalcopyrite dissolution. In the initial period (80 min of oxidation), this effect manifested itself more clearly then weakened, and approaching 230 min, the curves almost leveled out. The dissolution of chalcopyrite during 230 min of oxidation increased from 49.5 up to 50.9% with an increase in oxygen pressure from 0.2 up to 0.8 MPa.

The initial concentration of sulfuric acid had a pronounced negative effect on chalcopyrite dissolution (Figure 3b). Increasing the initial acid concentration from 10 up to 50 g/L led to a decrease in the degree of chalcopyrite dissolution from 59.9 down to 49.1% in 230 min.

The initial concentration of copper (II) ions had a positive effect on chalcopyrite dissolution throughout oxidation (Figure 3c). Its increase from 1 up to 2 g/L helped to increase the degree of chalcopyrite dissolution from 50 up to 53% within 230 min; a further increase in concentration up to 3 g/L had almost no effect.

The positive effect of the initial concentration of iron (III) ions was clearly manifested within 90 min of the process. Its increase from 2 up to 7 g/L helped to raise the degree of chalcopyrite dissolution from 43.5 up to 47.2%; a further increase in concentration up to 10 g/L had almost no effect. The possible reason for the increase in dissolution kinetics during 60–70 min is that the Fe (III) ions are an additional oxidizing agent.

After 90 min of the process, the curves began to level out, and after 170 min, iron (III) ions had a negative effect on chalcopyrite dissolution.

The negative effect of increasing the initial concentration of sulfuric acid may be associated with an increased oxidation degree of sulfide sulfur to elemental sulfur, which, in turn, screens the surface of chalcopyrite. This is also indicated by a change during the curves at the end of the process with an increase in the concentration of iron (III) ions, which are known to promote sulfide sulfur oxidation to the elemental state [32].

The X-ray phase analysis of the cake after oxidation of chalcopyrite is visualized in Figure 4.

According to the data presented in Figure 4, in addition to under-oxidized chalcopyrite, the cake contained elemental sulfur, whose content was 16.4%.

Microphotographs and EDX mapping for the cake obtained at t = 100 °C, P_o2_ = 0.8 MPa, [H_2_SO_4_] = 50 g/L, [Cu^2+^] = 3 g/L, [Fe^3+^] = 10 g/L, duration 230 min, are presented in Figure 5.

According to Figure 5a,b, chalcopyrite particles after low-temperature pressure leaching may have both smooth and loose, heterogeneous surfaces. The red zones in Figure 5d correspond to the distribution of sulfur, the green zones are responsible for iron and the yellow zones are for copper, respectively. A mixture of these zones is represented by chalcopyrite. It is noticeable from Figure 5c that elemental sulfur covers the surface of chalcopyrite, the particles being bright red.

In the experiment under these conditions, the oxidation degree of chalcopyrite with the formation of elemental sulfur and sulfate ions was 37.8% and 7.7%, respectively.

Based on the obtained microphotographs and EDX mappings presented in Figure 5, as well as the influence of increased initial concentrations of sulfuric acid and iron ions on the chalcopyrite dissolution degree, we can conclude that during low-temperature pressure oxidation, the surface is passivated by an elemental sulfur film, thereby limiting the access of reagents to the reaction zone, which is also confirmed by other researchers [32].

### 3.2. Dissolution of Pyrite

The interaction of pyrite with sulfuric acid in the presence of oxygen and iron (III) ions may proceed through the following reactions:1.25FeS_2_ + H_2_SO_4_ + O_2_ = 1.25FeSO_4_ + 2.25S^0^ + H_2_O ∆G (100 °C) = −368.076 kJ/mol(12)
FeS_2_ + 3.5O_2_ + H_2_O = FeSO_4_ + H_2_SO_4_ ∆G (100 °C) = −1075.317 kJ/mol(13)
2FeS_2_ + 4.5O_2_(g) + H_2_SO_4_ = Fe_2_(SO_4_)_3_ + 2S^0^ + H_2_O ∆G (100 °C) = −1435.262 kJ/mol(14)
1.75FeS_2_ + Fe_2_(SO_4_)_3_ + H_2_SO_4_ = 3.75FeSO_4_ + 3.75S^0^ + H_2_O ∆G (100 °C) = −192.382 kJ/mol(15)
4FeSO_4_ + O_2_(g) + 2H_2_SO_4_ = 2Fe_2_(SO_4_)_3_ + 2H_2_O ∆G (100 °C) = −304.494 kJ/mol(16)

According to the reactions presented, pyrite sulfide sulfur may be oxidized by oxygen to form elemental sulfur and sulfate ions (Reactions (12)–(14)). Iron (III) ions may also act as an oxidizing agent, with sulfide sulfur becoming an elemental one (Reaction (15)), and iron (II) ions react with oxygen to form iron (III) ions (Reaction (16)).

The influence of the oxygen partial pressure; the initial concentration of sulfuric acid, copper (II) and iron (III) ions; and the duration of pyrite dissolution are presented in Figure 6.

According to Figure 6a, increasing the oxygen pressure from 0.2 up to 0.8 MPa had a significant effect on pyrite dissolution in contrast to chalcopyrite (cf. Figure 3a). The positive effect was evident throughout the pressure low-temperature oxidation. Pyrite dissolution within 230 min of the process increased from 43.4 up to 51.1% with an increase in oxygen pressure from 0.2 up to 0.8 MPa.

The initial concentration of sulfuric acid had a pronounced positive effect on pyrite dissolution (Figure 6b). An increase in the initial acid concentration from 10 up to 50 g/L led to an increase in the pyrite dissolution degree from 45.1 up to 54.5% for 230 min. The positive effect of sulfuric acid may be associated with a decreased formation of iron hydroxides and oxides on the surface of pyrite particles capable of shielding its surface, complicating access to the reagents [33].

The initial copper (II) ion concentration had a small positive effect on pyrite dissolution throughout oxidation (Figure 6c). Its increase from 1 up to 2 g/L helped to increase the pyrite dissolution degree from 48.4% up to 51.1% within 230 min; a further increase in concentration up to 3 g/L had almost no effect.

Iron (III) ions also had a small positive effect on pyrite oxidation. Increasing the concentration from 2 up to 10 g/L helped to increase the pyrite dissolution degree from 47.4 up to 52.1%.

The X-ray phase analysis of the cake after chalcopyrite oxidation is visualized in Figure 7.

According to the data presented in Figure 7, in addition to under-oxidized pyrite, the cake contained elemental sulfur, whose content was 13.3%.

Microphotographs and EDX mapping for the cake obtained at t = 100 °C, P_o2_ = 0.8 MPa, [H_2_SO_4_] = 50 g/L, [Cu^2+^] = 3 g/L, [Fe^3+^] = 10 g/L, duration 230 min, are presented in Figure 8.

According to Figure 8a–c, pyrite particles after low-temperature pressure leaching have both smooth and loose heterogeneous surfaces. Small growths were found on the surface of rectangular particles (Figure 8b). The red zones in Figure 8d correspond to the distribution of sulfur and the green zones are responsible for iron, respectively. A mixture of these zones is represented by pyrite. It is noticeable from Figure 8f that elemental sulfur covers the surface of pyrite, its growths being bright red. This is especially true on grains with a more uneven surface. Fine particles are almost entirely elemental sulfur. On large grains with a smooth surface, elemental sulfur forms in uneven areas.

In the experiment under these conditions, the pyrite oxidation degree with the formation of elemental sulfur and sulfate ions was 17.4% and 37.3%, respectively. Unlike chalcopyrite, pyrite under identical conditions dissolves predominantly with the oxidation of sulfide sulfur to sulfate ions.

Based on the obtained microphotographs and EDX mappings presented in Figure 3, as well as our experimental results, we can conclude that the pyrite surface is passivated by an elemental sulfur film during low-temperature pressure oxidation, thereby limiting the access of the reagents to the reaction zone, which is also confirmed by other researchers [33].

### 3.3. Dissolution of a 1:1 Pyrite:Chalcopyrite Mixture

After studying the behavior of pyrite and chalcopyrite during low-temperature pressure oxidation, studies were made on their 1:1 mixture to evaluate the influence of these minerals on each other under identical conditions. The oxidation degree of CuFeS_2_ and FeS_2_ was assessed through the extraction degree of copper from chalcopyrite and iron from pyrite, respectively.

#### 3.3.1. Dissolution of Chalcopyrite

The interaction of chalcopyrite with sulfuric acid in the presence of oxygen and iron (III) ions may proceed according to the reactions described above (Equations (7)–(10)).

The influence of oxygen partial pressure; the initial concentration of sulfuric acid, copper (II) and iron (III) ions; and the duration on chalcopyrite dissolution are presented in Figure 9.

According to Figure 2a, increasing the oxygen pressure from 0.2 up to 0.8 MPa, as in the case of dissolving chalcopyrite alone, had a minor effect. The dissolution of chalcopyrite during 230 min of oxidation rose from 49.2 up to 50.9% with an increase in oxygen pressure from 0.2 up to 0.8 MPa.

The initial concentration of sulfuric acid during the dissolution of chalcopyrite alone had a pronounced negative effect (Figure 3b). The addition of pyrite in a 1:1 ratio contributed to a change in the nature of the relationship between the oxidation degree of chalcopyrite and the initial acid concentration (Figure 9b). The positive effect was observed throughout the entire process. Increasing the initial acid concentration from 10 up to 50 g/L led to an increased degree of chalcopyrite dissolution from 36.9 up to 50.1% in 230 min.

The initial concentration of copper (II) ions, as in the case of an individual chalcopyrite dissolution, had a positive effect on oxidation (Figure 9c). Its increase from 1 up to 3 g/L helped to increase the degree of chalcopyrite dissolution from 41.3 up to 47.7% within 230 min.

When chalcopyrite monosulfide was dissolved, an increase in the initial concentration of iron (III) ions had a positive effect within 170 min, and then (up to 230 min) negatively affected its oxidation (Figure 3d). When pyrite was added in a 1:1 ratio, an increase in the iron (III) ion concentration had a negative effect on the oxidation of chalcopyrite throughout the process (Figure 9d). Increasing their concentration from 2 up to 10 g/L led to a decreased degree of chalcopyrite dissolution from 49.1 to 43.1% in 230 min.

The negative effect of increasing the initial concentration of iron (III) ions may be associated with an increased oxidation degree of sulfide sulfur to elemental one, which, in turn, screens the surface of chalcopyrite [32].

In general, the maximum degree of chalcopyrite dissolution in its mixture with pyrite did not differ from the oxidation of the individual sulfide.

#### 3.3.2. Dissolution of Pyrite

The interaction of pyrite with sulfuric acid in the presence of oxygen and iron (III) ions may proceed according to the reactions described above (Equations (12)–(15)).

The influence of the partial pressure of oxygen; the initial concentration of sulfuric acid, copper (II) and iron (III) ions; and the duration of pyrite dissolution are presented in Figure 10.

Increasing the oxygen partial pressure from 0.2 up to 0.8 MPa also had a significant positive effect on pyrite dissolution in its mixture with chalcopyrite, as in the oxidation of its monosulfide (Figure 10a and Figure 6a). The positive effect was evident throughout the entire pressure low-temperature oxidation. Pyrite dissolution within 230 min of the process increased from 64.2 up to 79.4% with an increase in oxygen pressure from 0.2 up to 0.8 MPa.

The initial concentration of sulfuric acid had a pronounced positive effect on pyrite dissolution both in its mixture with chalcopyrite and without chalcopyrite addition (Figure 10b and Figure 6b). Increasing the initial acid concentration from 10 up to 50 g/L led to an increased degree of pyrite dissolution from 70.5 up to 80.3% in 230 min. The positive effect of sulfuric acid is also possibly associated with a decreased formation of iron hydroxides and oxides on the surface of pyrite particles capable of shielding its surface, complicating access to the reagents [33].

The initial concentration of copper (II) ions had a small positive effect on pyrite dissolution in its mixture with chalcopyrite throughout oxidation (Figure 10c). When its monosulfide was leached, the same effect was observed (Figure 6c). Its increase from 1 up to 2 g/L helped to raise the pyrite dissolution degree from 74.9 up to 78.5% within 230 min; a further increase in concentration up to 3 g/L had almost no effect.

Iron (III) ions also had a small positive effect on the oxidation of pyrite mixed with chalcopyrite throughout oxidation (Figure 10d). When its monosulfide was leached, the same effect was observed (Figure 6d). An increase in the concentration of iron (III) ions from 2 up to 7 g/L promoted an increased degree of pyrite dissolution from 74.6 up to 78.9% within 230 min; a further increase in concentration up to 10 g/L had virtually no effect.

In contrast to chalcopyrite dissolution, the pyrite oxidation degree in the mixture increased significantly. Under the following conditions: t = 100 °C, P_O2_ = 0.8 MPa, [H_2_SO_4_] = 50 g/L, [Cu^2+^] = 3 g/L, [Fe^3+^] = 10 g/L, the oxidation state of pyrite in its mixture with chalcopyrite increased from 54.5 up to 80.3% relative to the dissolution of the individual mineral.

#### 3.3.3. Analysis of the Cake of Low-Temperature Dissolution of a 1:1 Chalcopyrite:Pyrite Mixture

X-ray phase analysis of the cake after low-temperature pressure oxidation of a 1:1 chalcopyrite:pyrite mixture is visualized in Figure 11.

According to the data presented in Figure 11, in addition to under-oxidized pyrite and chalcopyrite, the cake contained elemental sulfur, whose content was 19.2%. It can also be noted that the pyrite peaks in the X-ray diffraction pattern have much lower intensity than the chalcopyrite ones.

Microphotographs and EDX mapping for the cake obtained at t = 100 °C, Po_2_ = 0.8 MPa, [H_2_SO_4_] = 50 g/L, [Cu^2+^] = 3 g/L, [Fe^3+^] = 10 g/L, duration 230 min of the oxidation of a 1:1 chalcopyrite:pyrite mixture are presented in Figure 12.

According to Figure 12a–c, the cake is represented by different forms of particles with different shapes. The first form, pyrite, has a smooth surface in places with grooves and cavities, corroded during the oxidation reaction. The second form, chalcopyrite, is represented by particles with a heterogeneous surface with pronounced defects whose inclusions of various shapes have formed (Figure 12c). The particle size of these cakes is 100% < 50 μm.

The red zones in Figure 12d correspond to sulfur distribution, the green zones are responsible for iron and the yellow zones are for copper, respectively. Elemental sulfur was distributed more evenly when a pyrite:chalcopyrite mixture was leached than when they were oxidized separately, especially for pyrite (Figure 12d). There are no incrustations of elemental sulfur thereon, which were found during dissolution with no addition of chalcopyrite, even in places with a more developed surface.

On the surface of chalcopyrite, on the contrary, there are incrustations upon which sulfur is unevenly distributed, and, according to Figure 12d, they contain elemental sulfur inclusions. In general, sulfur on the surface is also distributed more evenly than when dissolving without adding pyrite.

A microphotograph of the cake after pressure low-temperature oxidation of a chalcopyrite:pyrite mixture with composition analysis points is shown in Figure 13, and the element contents are presented in Table 2.

The data obtained confirm that the amount of elemental sulfur is about 0.1% on the surface of pyrite particles (Figure 13, Points 1 and 2), while its content reaches 5.9–8.3% on chalcopyrite particles.

In the experiment under these conditions, the degree of oxidation of sulfide sulfur to elemental one and sulfate ions was 23.2% and 46.5%, respectively.

According to previously obtained data on the behavior of monosulfides during low-temperature pressure oxidation, it can be concluded that an increase in the degree of pyrite dissolution with the addition of chalcopyrite is not associated with increased concentrations of copper (II) and iron (III) ions during oxidation since their influence on the degree of opening of minerals (including their mixtures) was insignificant. The positive effect on the pyrite oxidation degree in its mixture with chalcopyrite can be explained by the formation of an electrochemical link between the minerals, which has been widely described by other researchers [34,35,36,37,38,39,40]. This is also indicated by the change in the influence of the initial concentration of sulfuric acid and iron (III) ions on the oxidation degree of chalcopyrite in its mixture with pyrite (Figure 3 and Figure 9).

The potential for the same sulfide varies significantly depending on the nature of the solution and the degree of polarization. Therefore, the order of oxidation and dissolution of sulfides may also vary depending on various conditions. But the nature of oxidation and dissolution changes significantly when various sulfides are oxidized together. When an electrochemical couple is formed between pyrite and chalcopyrite in the presence of strong oxidizing agents (oxygen, iron (III) ions), dissolution is limited by electron transfer through the passivating film between the minerals [36]; therefore, unlike the Galvanox™ process, pyrite is oxidized as well.

The positive effect of chalcopyrite addition may be associated with a decreased formation of elemental sulfur on the pyrite surface, which is confirmed by microphotographs and EDX mapping data. This is also indicated by the fact that the highest rate of pyrite and chalcopyrite dissolution was observed within 50 min of oxidation. Further, chalcopyrite dissolution slowed down sharply due to the passivation of its surface with elemental sulfur; the nature of the curves is identical to oxidation with no addition of pyrite. At the same time, the rate of pyrite oxidation after 50 min of the process also began to slow down, but not as sharply as in the case of oxidation with no addition of chalcopyrite. Apparently, during this period, its surface also begins to be passivated with elemental sulfur.

## 4. Conclusions

In this study, we conducted experiments on the low-temperature pressure oxidation of pyrite, chalcopyrite individually and the mixture of both materials. The series of experiments on chalcopyrite dissolution showed that increasing the oxygen pressure from 0.2 to 0.8 MPa had a minimal effect on chalcopyrite dissolution, while an increase in the initial concentration of sulfuric acid had a significant negative impact. Specifically, raising the initial acid concentration from 10 to 50 g/L resulted in a decrease in the degree of chalcopyrite dissolution from 59.9% to 49.1% over 230 min. The obtained SEM and EDX mappings indicated that the surface was passivated by elemental sulfur.

The degree of pyrite oxidation was most positively influenced by the oxygen pressure and the initial concentration of sulfuric acid. Increasing the oxygen pressure from 0.2 to 0.8 MPa increased pyrite dissolution from 43.4% to 51.1%, while raising the initial concentration of sulfuric acid from 10 to 50 g/L enhanced its oxidation degree from 45.1% to 54.5%. The SEM and EDX mappings also indicated that the pyrite surface was passivated by elemental sulfur, particularly in areas with a developed surface.

When pyrite was mixed with chalcopyrite, the oxidation degree of pyrite significantly increased from 54.5% to 80.3% compared to the dissolution of the individual mineral, while the dissolution of chalcopyrite remained virtually unchanged. The increase in pyrite dissolution with the addition of chalcopyrite is not associated with increased concentrations of copper (II) and iron (III) ions during oxidation, as their effect on the degree of mineral exposure (including their mixtures) was insignificant. The positive effect on the oxidation degree of pyrite in its mixture with chalcopyrite can be explained by the formation of an electrochemical link between the minerals. This is further supported by a change in the influence of the initial concentration of sulfuric acid and iron (III) ions on the oxidation degree of chalcopyrite in its mixture with pyrite.

The positive effect of adding chalcopyrite is associated with a reduced formation of elemental sulfur on the surface of pyrite, as confirmed by SEM and EDX mapping data. The content of elemental sulfur on the surfaces of pyrite and chalcopyrite particles was approximately 0.1% and 5.9–8.3%, respectively. This is also reflected in the observation that the highest rates of dissolution for both pyrite and chalcopyrite occurred within the first 50 min of oxidation. Following this period, the dissolution of chalcopyrite sharply slowed due to the passivation of its surface with elemental sulfur, while the pyrite oxidation rate also began to decrease after 50 min, indicating that its surface was also starting to be passivated by elemental sulfur during this time.

This method can be applied to hydrometallurgical processing of copper raw materials with high contents of pyrite, and also for pretreatment of gold-bearing pyrite concentrates with copper concentrates to enhance the extraction rate of gold and silver.

## Figures and Tables

**Figure 1 materials-17-05132-f001:**
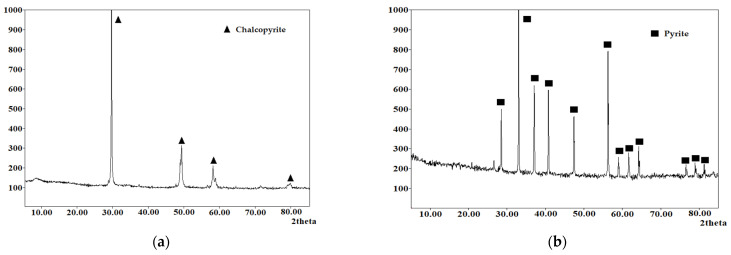
X-ray diffraction pattern of phase composition. (**a**) chalcopyrite; (**b**) pyrite.

**Figure 2 materials-17-05132-f002:**
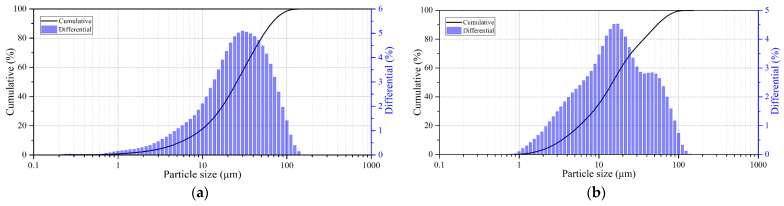
Granulometric composition of the sulfide minerals. (**a**) chalcopyrite; (**b**) pyrite.

**Figure 3 materials-17-05132-f003:**
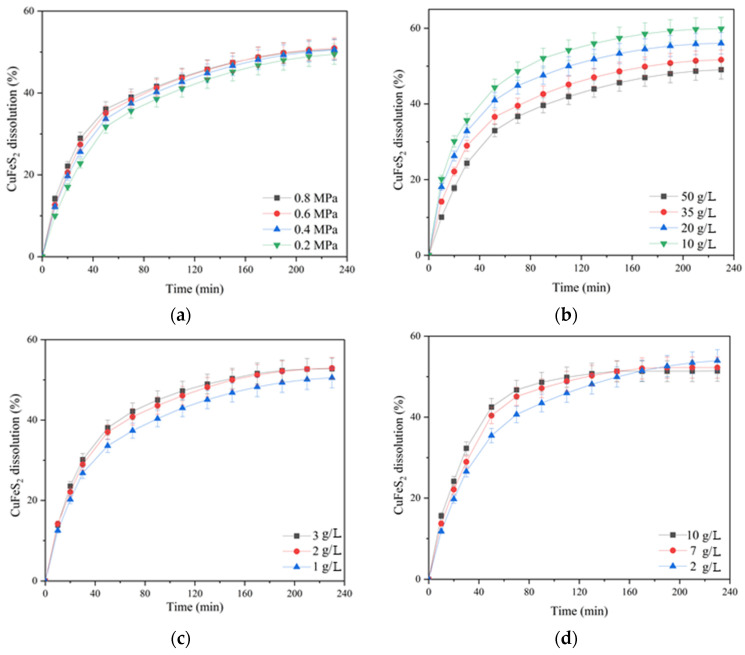
Dependence of chalcopyrite dissolution in sulfuric acid solutions in the presence of oxygen on the partial pressure of (**a**) oxygen and (**b**) the initial concentration of sulfuric acid, (**c**) copper (II) and (**d**) iron (III) ions.

**Figure 4 materials-17-05132-f004:**
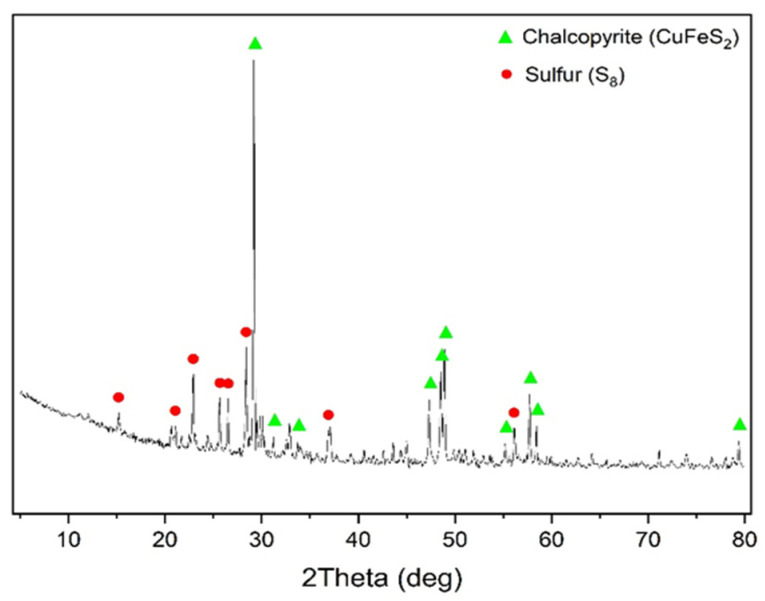
X-ray diffraction pattern of the cake of pressure low-temperature oxidation of chalcopyrite (t = 100 °C, P_o2_ = 0.8 MPa, [H_2_SO_4_] = 50 g/L, [Cu^2+^] = 3 g/L, [Fe^3+^] = 10 g/L, duration 230 min).

**Figure 5 materials-17-05132-f005:**
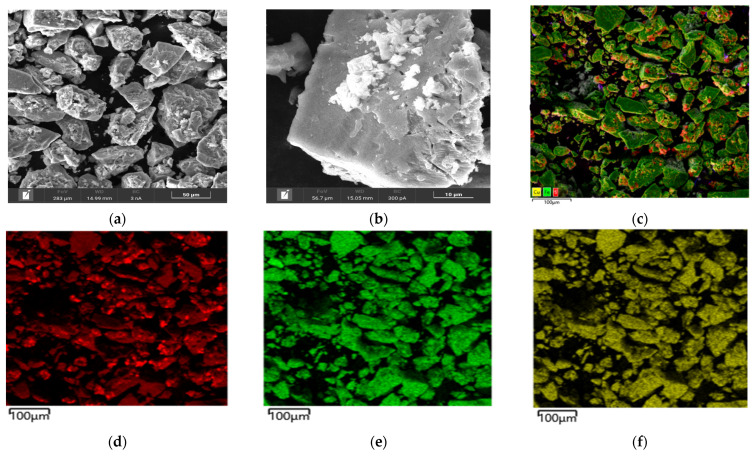
(**a**,**b**) SEM images of chalcopyrite oxidation cake particles and EDS mapping (**c**) for the mixture, (**d**) sulfur, (**e**) iron and (**f**) copper.

**Figure 6 materials-17-05132-f006:**
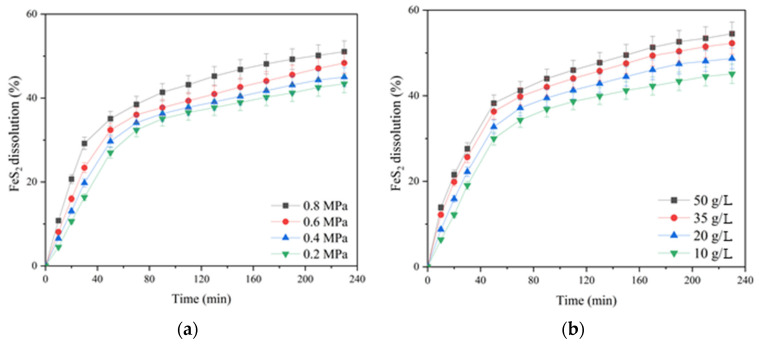
Dependence of the pyrite dissolution degree in sulfuric acid solutions in the presence of oxygen on (**a**) the partial pressure of oxygen and (**b**) the initial concentration of sulfuric acid, (**c**) copper (II) and (**d**) iron (III) ions.

**Figure 7 materials-17-05132-f007:**
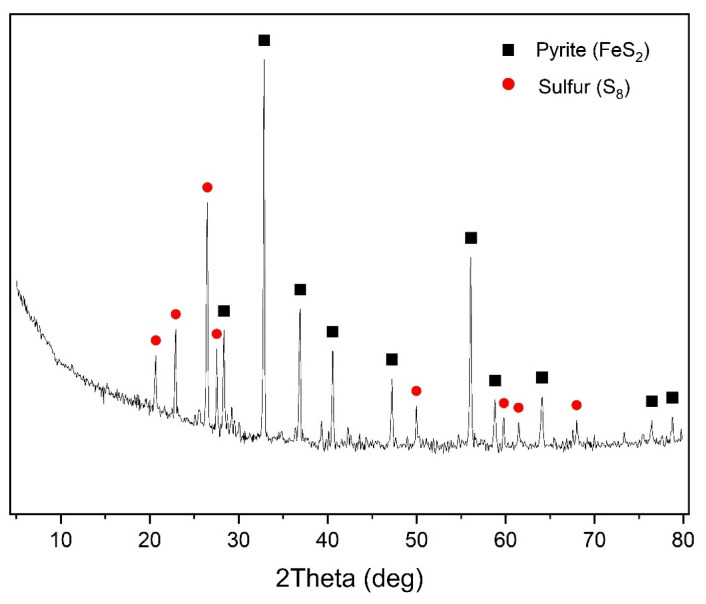
X-ray diffraction pattern of the cake of pressure low-temperature pyrite oxidation (t = 100 °C, Po_2_ = 0.8 MPa, [H_2_SO_4_] = 50 g/L, [Cu^2+^] = 3 g/L, [Fe^3+^] = 10 g/L, duration 230 min).

**Figure 8 materials-17-05132-f008:**
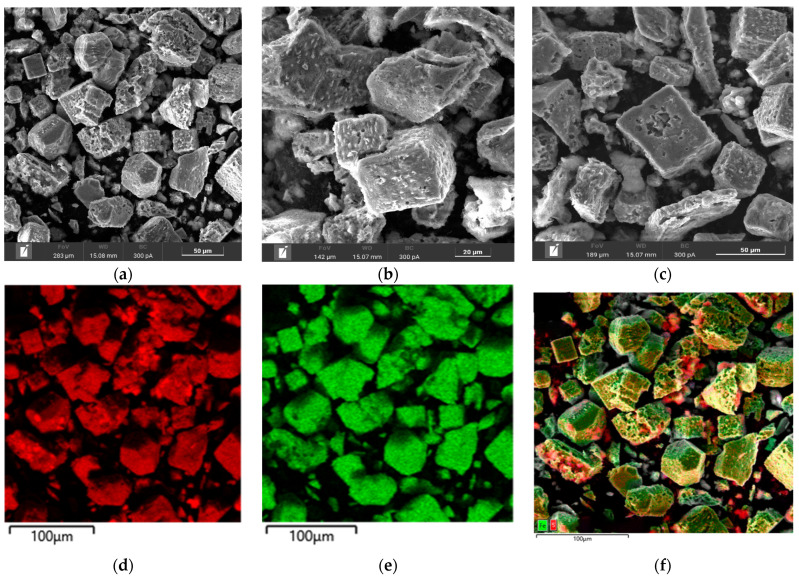
(**a**–**c**) SEM images of pyrite oxidation cake particles and EDS mapping for (**d**) sulfur, (**e**) iron and (**f**) their mixture.

**Figure 9 materials-17-05132-f009:**
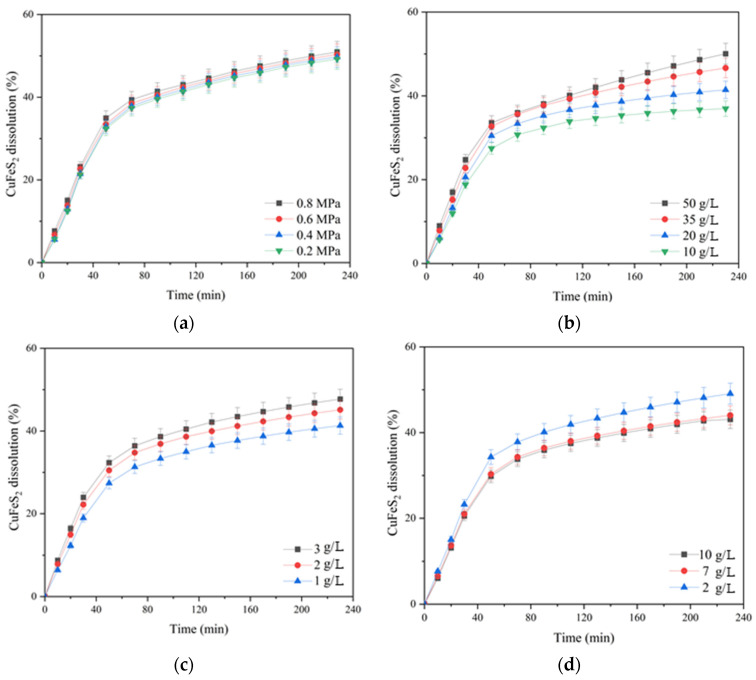
Dependence of the chalcopyrite dissolution in sulfuric acid solutions in a mixture with pyrite in the presence of oxygen on (**a**) the partial pressure of oxygen and (**b**) the initial concentration of sulfuric acid, (**c**) copper (II) and (**d**) iron (III) ions.

**Figure 10 materials-17-05132-f010:**
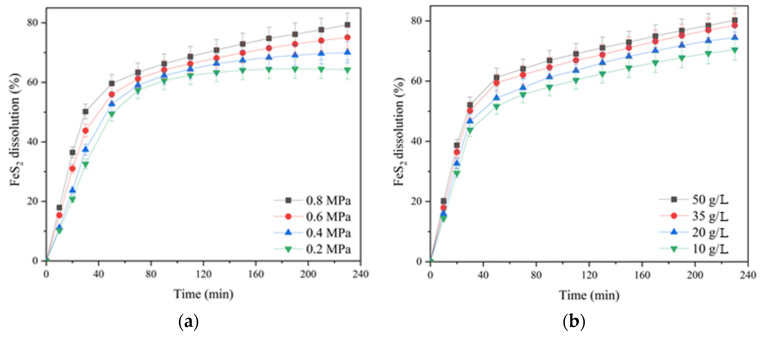
Dependence of pyrite dissolution in sulfuric acid solutions in a mixture with pyrite in the presence of oxygen on (**a**) the partial pressure of oxygen and (**b**) the initial concentration of sulfuric acid, (**c**) copper (II) and (**d**) iron (III) ions.

**Figure 11 materials-17-05132-f011:**
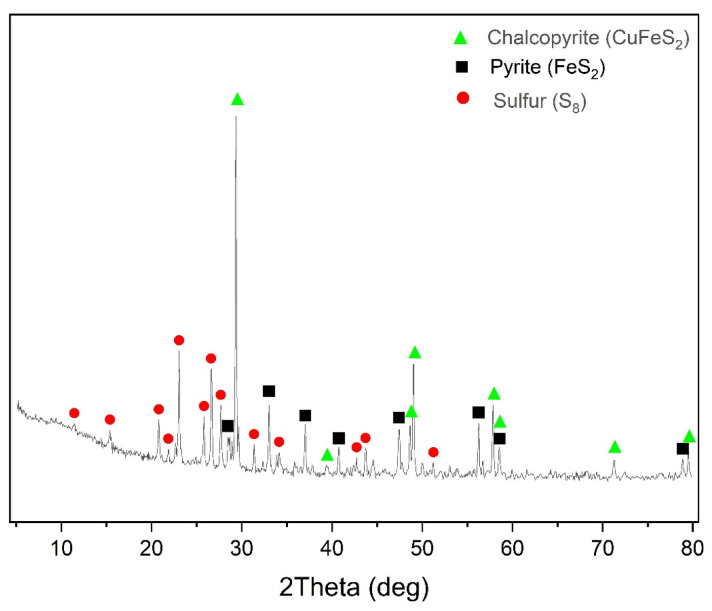
X-ray diffraction pattern of the cake of pressure low-temperature pyrite oxidation (t = 100 °C, Po_2_ = 0.8 MPa, [H_2_SO_4_] = 50 g/L, [Cu^2+^] = 3 g/L, [Fe^3+^] = 10 g/L, duration 230 min).

**Figure 12 materials-17-05132-f012:**
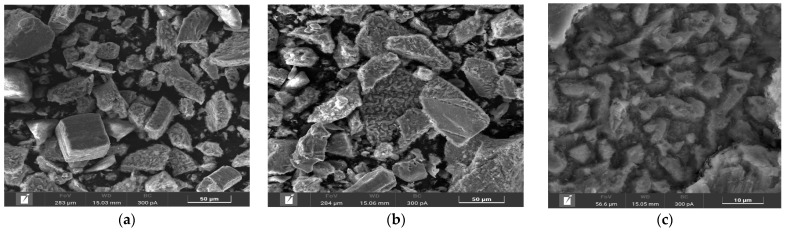
(**a**–**c**) SEM images of particles of the cake of oxidation of a 1:1 pyrite:chalcopyrite mixture and EDS mapping for (**d**) the mixture, (**e**) sulfur, (**f**) iron and (**g**) copper.

**Figure 13 materials-17-05132-f013:**
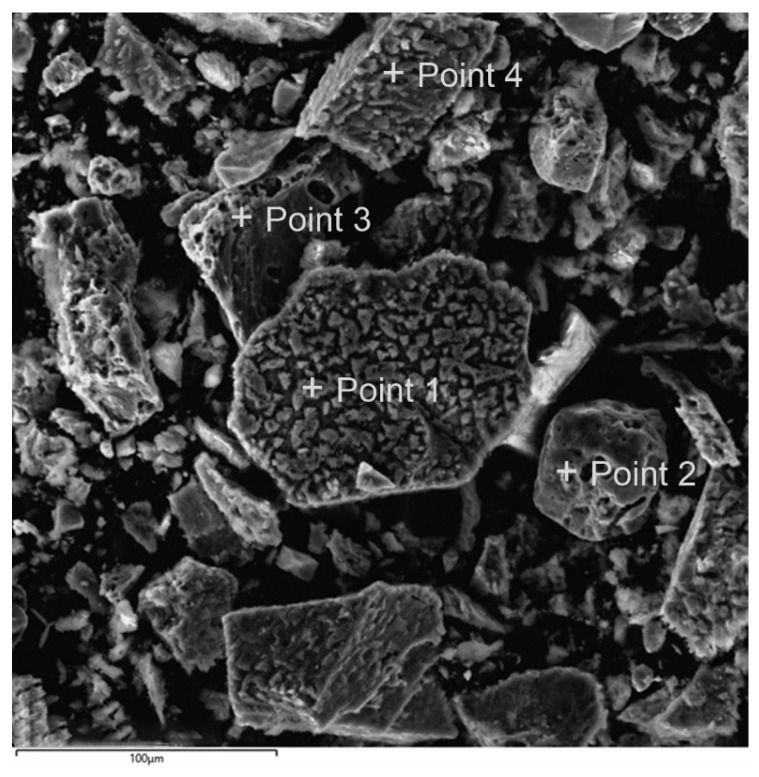
Microphotograph of the cake after pressure low-temperature oxidation of a chalcopyrite:pyrite mixture with points for analyzing its composition.

**Table 1 materials-17-05132-t001:** The chemical composition of the used minerals.

Materials	Content/wt.%			
	Cu	Fe	S	Others
Chalcopyrite	33.4	32.0	33.6	1.0
Pyrite		44.1	50.8	5.1

**Table 2 materials-17-05132-t002:** Contents of elements at the composition analysis points.

Element	Fe	Cu	S_sulfide_	S^0^	Total
Figure 13. Point 1	29.0	31.2	31.4	8.3	100
Figure 13. Point 2	46.1	0.4	53.3	0.1	100
Figure 13. Point 3	45.9	0.7	53.3	0.1	100
Figure 13. Point 4	30.0	32.0	32.2	5.9	100

## Data Availability

The original contributions presented in the study are included in the article, further inquiries can be directed to the corresponding author.

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
