# Peer review of "The Dissolution Behavior of Pyrite and Chalcopyrite During Low-Temperature Pressure Oxidation: Chalcopyrite Influence on Pyrite Oxidation"

_materials, 2024, doi:10.3390/ma17205132_

Round 1

Reviewer 1 Report

Comments and Suggestions for Authors

This paper investigates the low-temperature pressure oxidation of pyrite, chalcopyrite, and their mixture in sulfuric acid solutions. The study examines the effects of oxygen partial pressure, initial concentration of sulfuric acid, copper (II) and iron (III) ions, and duration on the dissolution of individual minerals and their mixture. The results show that the addition of chalcopyrite to pyrite increases the pyrite oxidation degree, while the dissolution of chalcopyrite remains unchanged. The study suggests that the formation of an electrochemical link between the minerals is responsible for the increased pyrite oxidation degree.

1. In the abstract, the sentence "The increase in the pyrite dissolution degree with the addition of chalcopyrite is not associated with increased concentrations of copper (II) and iron (III) ions during oxidation" is unclear.

2. In the introduction, the sentence "The main direction of development of hydrometallurgical methods is conducting the process at low temperatures in an autoclave or even under atmospheric pressure" is too broad. Specify the temperature range and pressure conditions.

3. In Table 1, the chemical composition of the minerals is not normalized to 100%.

4. In the experimental section, the particle size distribution of the minerals is not provided.

5. In Figure 3, the error bars are not provided. Include error bars to represent the uncertainty in the data.

6. In the discussion section, the sentence "The positive effect of chalcopyrite addition is associated with a decreased formation of elemental sulfur on the surface of pyrite" is not supported by the data. Provide evidence from the data to support this claim.

This paper requires a major revision to address the above comments. The authors need to provide more detailed information on the experimental conditions, data analysis, and discussion. The paper should be rewritten to ensure clarity, concision, and accuracy.

Comments on the Quality of English Language

Minor revision is required.

Author Response

Dear Reviewer! We thank you for your interest to our paper. We are glad to answer your comments about our work.

Comment 1. In the abstract, the sentence "The increase in the pyrite dissolution degree with the addition of chalcopyrite is not associated with increased concentrations of copper (II) and iron (III) ions during oxidation" is unclear.

Answer 1. It means that an addition of Cu (II) and Fe (III) ions does not influence on the pyrite dissolution when chalcopyrite was added in leaching process. It is corrected in text (line 18, 19).

Comment 2. In the introduction, the sentence "The main direction of development of hydrometallurgical methods is conducting the process at low temperatures in an autoclave or even under atmospheric pressure" is too broad. Specify the temperature range and pressure conditions.

Answer 2. The temperature range is 100 ± 2 °C and the pressure range
is 0 – 0,8 MPa. It is added in line 30, 31.

Comment 3. In Table 1, the chemical composition of the minerals is not normalized to 100%.

Answer 3. It is normalized to 100 %.

Comment 4. In the experimental section, the particle size distribution of the minerals is not provided.

Answer 4. The particle size distribution of minerals is presented in Figure 2, line 131.

Comment 5. In Figure 3, the error bars are not provided. Include error bars to represent the uncertainty in the data.

Answer 5. The Figure 3, 6, 9 and 10 were corrected including error bars.

Comment 6. In the discussion section, the sentence "The positive effect of chalcopyrite addition is associated with a decreased formation of elemental sulfur on the surface of pyrite" is not supported by the data. Provide evidence from the data to support this claim.

Answer 6. In this research many cake particles were analyzed with EDX method. This data was proven. We had written the average amount of sulfur for pyrite and chalcopyrite here.

Reviewer 2 Report

Comments and Suggestions for Authors

The Dissolution Behavior of Pyrite and Chalcopyrite during Low-Temperature Pressure Oxidation: The Chalcopyrite Influence on the Pyrite Oxidation is very important paper in hydrometallurgy of iron and copper. Minor improvements are required.

Line 10; In this work, studies were carried out on the low-temperature pressure oxidation of pyrite (in which temperature and pressure range?)

Line 13: The greatest positive effect on the pyrite oxidation degree was exerted by oxygen pressure and the initial concentration of sulfuric acid (In which concentration range?)

Line 15: The oxidation degree of pyrite  in its mixture with chalcopyrite increased significantly from 54.5 up to 80.3%  (in which reaction time?)

Line 60, 61, 62: It is assumed that the  type and structure of the films are greatly influenced by the type of mineral and leaching conditions: acid concentration, redox potential and process temperature (and reaction time?)

Line 92: which leads to an increased rate of the process ( what is maximal rate of process?)

Line 313: The initial concentration of sulfuric acid during the dissolution of chalcopyrite alone had a pronounced negative effect Figure 2b (or Figure 9b)

Line 390: According to Figure 12 (a,b,c), the cake is represented by two types of particles with different shapes (what are particle sizes?)

Author Response

Dear Reviewer! We thank you for your interest to our paper. We are glad to answer your comments about our work.

Comment 1. Line 10; In this work, studies were carried out on the low-temperature pressure oxidation of pyrite (in which temperature and pressure range?)

Answer 1. The temperature range is 100 ± 2 °C and the pressure range
is 0 – 0,8 MPa. It is added in line 30, 31.

Comment 2. Line 13: The greatest positive effect on the pyrite oxidation degree was exerted by oxygen pressure and the initial concentration of sulfuric acid (In which concentration range?)

Answer 2. The initial concentration range of sulfuric acid was 10–50 g/l. It is added in line 13,14.

Comment 3. Line 15: The oxidation degree of pyrite  in its mixture with chalcopyrite increased significantly from 54.5 up to 80.3%  (in which reaction time?)

Answer 3. The reaction time was 0–240 min. It is added in line 16.

Comment 4. Line 60, 61, 62: It is assumed that the  type and structure of the films are greatly influenced by the type of mineral and leaching conditions: acid concentration, redox potential and process temperature (and reaction time?)

Answer 4. We agree, it is “… and the reaction time”.

Comment 5. Line 92: which leads to an increased rate of the process ( what is maximal rate of process?)

Answer 5. It is one of the conclusions of  the literature’s data and the maximal diffusion rate of copper ions is not described. The link on papers [30,31] is in the text line 93.

Comment 6. Line 313: The initial concentration of sulfuric acid during the dissolution of chalcopyrite alone had a pronounced negative effect Figure 2b (or Figure 9b)

Answer 6. It is a misprint. The correct figure is “Figure 3b” (a link to an isolated leaching of chalcopyrite in line 183-185).

Comment 7. Line 390: According to Figure 12 (a,b,c), the cake is represented by two types of particles with different shapes (what are particle sizes?)

Answer 7. It was corrected to “different forms”. One is for pyrite, the other – for chalcopyrite. The particle size can be seen in SEM images (Figure 12 a,b,c) and it is 100 % < 50 μm. It is added in line 392-397.

Reviewer 3 Report

Comments and Suggestions for Authors

The manuscript titled “The Dissolution Behavior of Pyrite and Chalcopyrite during Low-Temperature Pressure Oxidation: The Chalcopyrite Influence on the Pyrite Oxidation" by Karimov et al. has investigated the low-temperature pressure oxidation of pyrite and chalcopyrite, both separately and as a mixture.

The authors should consider the following important points during the revision of this paper.

1.     Line 185-192: “According to Figure 3 (a), increasing the oxygen pressure from 0.2 up to 0.8 MPa had little effect on chalcopyrite dissolution.” Instead in Figure 3, it is presented in Figure 3b. “The initial concentration of sulfuric acid had a pronounced negative effect on chalcopyrite dissolution (Figure 3 (b).” Instead in Figure 3a.

2.     Figure 3 caption must be corrected accordingly.

3.     The units in figures are presented in g/l, while in text it is mentioned in g/dm3. Kindly follow consistent units throughout.

4.     Line 194-197: The data presented in Figure 3c do not have clear error margins or statistical analysis, so based on the present data it is difficult to conclude that increase in the degree of chalcopyrite dissolution 50% and 53% may fall within error margins. Authors must include data presented with error margins.

5.     Line 199-200: Kindly discuss the possible reason for the increase in dissolution kinetics during 60-70 minute, in presence of iron(III) and also include the data with error margins.

6.     Line 258-263: Copper concentration should be 1g/l to 3 g/l (Figure 6c), while iron concentration should be 2 g/l to 10 g/l.

7.     The results presented here for the sulfur content on the surface of chalcopyrite and pyrite is based on EDX data, but the exact amounts like 0.1% for pyrite and 5.9–8.3% for chalcopyrite, do not seem precise, especially without clear error margins or statistical analysis.

8.     More detailed discussion on why chalcopyrite addition reduces elemental sulfur formation on pyrite, may be helpful.

9.     Kindly discuss the electrochemical processes that may occur during the oxidation of the mineral mixture.

10.  What is the potential role of galvanic interactions between the minerals, which could be supported by electrochemical measurements or references to existing studies, must be included.

11.  Authors should discuss in detail the implications of these findings. How can this research contribute to improving industrial processes, particularly in the context of pyrometallurgical and hydrometallurgical applications?

12.  Kindly also discuss how the reduced sulfur passivation improve overall metal recovery, and what are the potential benefits for the mining and metal extraction industries.

13.  Conclusion must be revised with focus on the broader significance and implications of the findings, particularly in the context of industrial applications. Avoid repeating results.

14.  The text need editing, grammatical errors and awkward sentences should be improved. For example, "In this work, studies were carried out on the low-temperature pressure oxidation of pyrite, chalcopyrite separately, and their mixture."; "The greatest positive effect on the pyrite oxidation degree was exerted by oxygen pressure” etc.

Comments on the Quality of English Language

Language editing required.

Author Response

Dear Reviewer! We thank you for your interest to our paper. We are glad to answer your comments about our work.

Comment 1. Line 185-192: “According to Figure 3 (a), increasing the oxygen pressure from 0.2 up to 0.8 MPa had little effect on chalcopyrite dissolution.” Instead in Figure 3, it is presented in Figure 3b. “The initial concentration of sulfuric acid had a pronounced negative effect on chalcopyrite dissolution (Figure 3 (b).” Instead in Figure 3a.

Answer 1.It is corrected.

Comment 2. Figure 3 caption must be corrected accordingly.

Answer 2. It is corrected.

Comment 3. The units in figures are presented in g/l, while in text it is mentioned in g/dm3. Kindly follow consistent units throughout.

Answer 3. The units are replaces from g/dm3 to g/l through the whole text.

Comment 4. Line 194-197: The data presented in Figure 3c do not have clear error margins or statistical analysis, so based on the present data it is difficult to conclude that increase in the degree of chalcopyrite dissolution 50% and 53% may fall within error margins. Authors must include data presented with error margins.

Answer 4. It is corrected.

Comment 5. Line 199-200: Kindly discuss the possible reason for the increase in dissolution kinetics during 60-70 minute, in presence of iron(III) and also include the data with error margins.

Answer 5. Data is corrected. The possible reason for the increase in dissolution kinetics during 60-70 minute is that the Fe (III) ions are an additional oxidizing agent (line 202, 203).

Comment 6. Line 258-263: Copper concentration should be 1g/l to 3 g/l (Figure 6c), while iron concentration should be 2 g/l to 10 g/l.

Answer 6. Copper concentration is written correctly. In line 261-264 it is described that an increase of Cu (II) ions from 1 to 2 g/L had a positive effect on pyrite leaching, but further increasing from 2 to 3 g/L had almost no effect.

Comment 7. The results presented here for the sulfur content on the surface of chalcopyrite and pyrite is based on EDX data, but the exact amounts like 0.1% for pyrite and 5.9–8.3% for chalcopyrite, do not seem precise, especially without clear error margins or statistical analysis.

Answer 7. In this research many cake particles were analyzed with EDX method. This data was proven. We had written the average amount of sulfur for pyrite and chalcopyrite here.

Comment 8. More detailed discussion on why chalcopyrite addition reduces elemental sulfur formation on pyrite, may be helpful.

Comment 9. Kindly discuss the electrochemical processes that may occur during the oxidation of the mineral mixture.

Comment 10. What is the potential role of galvanic interactions between the minerals, which could be supported by electrochemical measurements or references to existing studies, must be included.

Answer 8, 9,10. As it is fully described in [23-25] with graphs, figures and reactions, chalcopyrite is a semiconductor and therefore corrodes electrochemically in oxidizing solutions. Pyrite is an effective and convenient provider of this alternative surface for ferric reduction. The conclusion that “chalcopyrite addition reduces elemental sulfur formation on pyrite” is described in many works of different authors, for example, the Galvanox process [25] is in the text line 73-89.

Comment 11. Authors should discuss in detail the implications of these findings. How can this research contribute to improving industrial processes, particularly in the context of pyrometallurgical and hydrometallurgical applications?

Comment 12. Kindly also discuss how the reduced sulfur passivation improve overall metal recovery, and what are the potential benefits for the mining and metal extraction industries.

Answer 11, 12. This method can be used for hydrometallurgical processing of copper raw materials with increased pyrite content, as well as for pretreatment of copper concentrates with gold-rich pyrite concentrates to increase the recovery of gold and silver (line 21-24).

Comment 13. Conclusion must be revised with focus on the broader significance and implications of the findings, particularly in the context of industrial applications. Avoid repeating results.

Answer 13. We agree. It is fully corrected.

Comment 14. The text need editing, grammatical errors and awkward sentences should be improved. For example, "In this work, studies were carried out on the low-temperature pressure oxidation of pyrite, chalcopyrite separately, and their mixture."; "The greatest positive effect on the pyrite oxidation degree was exerted by oxygen pressure” etc.

Answer 14. It is corrected.

Round 2

Reviewer 3 Report

Comments and Suggestions for Authors

Significant improvement has been done in the revised manuscript. It can be recommended for publication.